# Non-Coding RNAs in Kidney Diseases: The Long and Short of Them

**DOI:** 10.3390/ijms22116077

**Published:** 2021-06-04

**Authors:** Juan Antonio Moreno, Eya Hamza, Melania Guerrero-Hue, Sandra Rayego-Mateos, Cristina García-Caballero, Mercedes Vallejo-Mudarra, Laurent Metzinger, Valérie Metzinger-Le Meuth

**Affiliations:** 1Maimonides Biomedical Research Institute of Cordoba (IMIBIC), UGC Nephrology, Hospital Universitario Reina Sofía, 14004 Córdoba, Spain; Melania.guerrero@imibic.org (M.G.-H.); Cristina.garcia@imibic.org (C.G.-C.); b52vamum@uco.es (M.V.-M.); 2Biomedical Research Networking Center on Cardiovascular Diseases (CIBERCV), 28029 Madrid, Spain; 3Department of Cell Biology, Physiology and Immunology, University of Cordoba, 140471 Cordoba, Spain; 4HEMATIM UR 4666, C.U.R.S, Université de Picardie Jules Verne (UPJV), CEDEX 1, 80025 Amiens, France; eya.hamza@etud.u-picardie.fr (E.H.); valerie.metzinger@univ-paris13.fr (V.M.-L.M.); 5Renal, Vascular and Diabetes Research Laboratory, IIS-Fundación Jiménez Díaz, Universidad Autónoma de Madrid, 28040 Madrid, Spain; srayego@quironsalud.es; 6INSERM UMRS 1148, Laboratory for Vascular Translational Science (LVTS), UFR SMBH, Université Sorbonne Paris Nord, CEDEX, 93017 Bobigny, France

**Keywords:** non-coding RNA, microRNA, long non-coding RNA, chronic kidney disease, acute kidney disease, IgA nephropathy, gene regulation

## Abstract

Recent progress in genomic research has highlighted the genome to be much more transcribed than expected. The formerly so-called junk DNA encodes a miscellaneous group of largely unknown RNA transcripts, which contain the long non-coding RNAs (lncRNAs) family. lncRNAs are instrumental in gene regulation. Moreover, understanding their biological roles in the physiopathology of many diseases, including renal, is a new challenge. lncRNAs regulate the effects of microRNAs (miRNA) on mRNA expression. Understanding the complex crosstalk between lncRNA–miRNA–mRNA is one of the main challenges of modern molecular biology. This review aims to summarize the role of lncRNA on kidney diseases, the molecular mechanisms involved, and their function as emerging prognostic biomarkers for both acute and chronic kidney diseases. Finally, we will also outline new therapeutic opportunities to diminish renal injury by targeting lncRNA with antisense oligonucleotides.

## 1. Introduction

In 1986, Walter Gilbert suggested that self-replicating RNA molecules were at the very origin of life, even before the appearance of DNA and proteins, and introduced the concept of the RNA world [1,2]. From that time onwards, many RNA species have been described. Deep sequencing technologies have revealed that most of the eukaryotic genome is transcribed and the existence of numerous non-coding RNAs (ncRNAs). ncRNAs are numerous and highly adapted in their roles in modern organisms, as RNA molecules are well designed to specifically recognize other RNAs and DNA by complementary base pairing. This complex network of transcripts include thousands of ncRNAs with little or no protein-coding capacity. They function as post-transcriptional regulatory molecules, interacting with specific mRNAs and determining protein expression. In the non-coding transcriptome, secondary and tertiary structures are crucial for their interaction with proteins or other nucleic acids. These recent findings extend beyond the DNA–RNA–protein central dogma coined by Francis Crick in 1957 [3].

In the last two decades, the small microRNAs (miRNAs) have arisen as important regulators of most pathophysiological mechanisms [4,5,6]. miRNAs are around 20–25 nucleotides long, single stranded RNAs. More than 2000 miRNAs are encoded by the human genome. During miRNA biogenesis the double stranded miRNA-3p/miRNA-5p duplex is unwound to single strands by the *RNA-induced silencing* *complex* (*RISC*) that presents the mature miRNA to its target mRNAs, triggering gene silencing in a post-transcriptional manner. The analysis of miRNAs in biofluids, including serum and plasma, has been proposed as a minimally invasive diagnostic approach to detect changes of the physiological status of patients.

Alongside the short miRNAs, a long list of bigger transcripts called long non-coding RNAs (lncRNAs) has been recently described. Interestingly, the majority of lncRNAs are associated with human diseases, much like miRNAs, but unlike their smaller counterparts, they display cell type-specific expression and localization to subcellular compartments and are much less conserved. Physiologically, lncRNAs have been shown to modulate gene regulation at all levels (including but not limited to promoter activity, epigenetics, translation and transcription efficiency, intracellular trafficking) [7]. As in the case with other tissues, the kidney expresses tens of thousands of lncRNAs sequences that are often conserved with coding genes. lncRNAs are divided into four categories: long intervening/intergenic non-coding RNAs, intronic lncRNAs, sense lncRNAs (including pseudogenes) and antisense lncRNAs [8]. Competitive binding between lncRNAs, target mRNAs and miRNAs are considered to regulate gene expression, forming a wide RNA-based transcriptional regulatory network [9]. The majority of lncRNAs are expressed in the nucleus, whereas a minority is cytoplasmic [10]. A promising feature of lncRNAs is that they are more tissue- or cell type-specific than mRNAs or miRNAs, suggesting a better biomarker potential.

As lncRNAs are implicated in all levels of the gene regulation process, we can find them deregulated in a variety of pathophysiological processes, particularly in diseases involving genomic imprinting and cancer [11].

Over recent years, lncRNAs have been increasingly recognized as being implicated in the pathophysiology of various kidney diseases. The aim of the present review is to examine the current literature regarding this field, the molecular mechanisms involved, and the value of lncRNAs as emerging prognostic biomarkers in renal diseases. Finally, we will also outline new therapeutic approaches targeting lncRNA to decrease renal damage.

## 2. Role of lncRNAs in Renal Diseases

### 2.1. Acute Kidney Injury

Acute kidney injury (AKI) is characterized by an acute loss of renal function that reduces urinary excretion of toxic substances. The accumulation of these waste products in blood can be lethal depending on the duration and severity of AKI, triggering the use of dialysis to ensure patient survival. The pathophysiology of AKI is complex, including a number of processes that promote direct tubular cell death, obstruction and activation of renal vessels and tubular lumen, exacerbated inflammatory response and oxidative stress. LncRNAs regulate numerous genes and mechanisms involved in the pathophysiology of AKI, serving both as biomarkers and therapeutical targets against renal damage. In the next section of the review, we will update recent findings about the role of lncRNAs in different AKI subtypes, including sepsis, ischemia reperfusion, drug-induced and obstructive subtypes (Table 1).

#### 2.1.1. Sepsis-Induced AKI

AKI may be induced in the context of severe sepsis [12]. In recent years, several studies have demonstrated the key role of lncRNAs in septic-induced AKI (SI-AKI). In a microarray study, 11.289 lncRNAs were differentially expressed in patients with SI-AKI, 5.361 were upregulated and 5.928 were downregulated. In this study, the lncRNA TCONS_00021712 was the most upregulated and TCONS_00016406 (lncRNA 6406) the most downregulated [13]. In line with these data, reduced renal lncRNA 6406 expression was observed in an experimental model of SI-AKI induced by administration of lipopolysaccharide (LPS) [14]. LncRNA 6406 decreased LPS-mediated inflammation, oxidative stress and cell death via regulation of miR-687/PTEN axis [14]. Furthermore, the lncRNA TCONS_00016233 plasma concentration was mainly elevated in sepsis-associated AKI patients, serving as an early diagnosis biomarker for this subtype of AKI [15]. In this study, lncRNA TCONS_00016233 expression was regulated by the TLR4 signaling pathway, and this lncRNA aggravated SI-AKI by targeting the miR-22-3p/AIFM1 pathway.

The lncRNA Metastasis Associated Lung Adenocarcinoma Transcript 1 (MALAT1) has been studied in SI-AKI. MALAT1 plasma levels have been shown to be elevated in the serum of patients with sepsis [16]. The expression of MALAT1 was increased in experimental LPS-induced AKI and LPS-treated renal cells, increasing renal injury and NF-κB activation throughout downregulation of miR-146a [17]. There are some compounds that reduce MALAT1 expression, such as dexmedetomidine and paclitaxel, resulting in the reduction of LPS injury in HK-2 cells [16,18]. Another lncRNA upregulated in septic patients with AKI is NEAT1 (Nuclear Paraspeckle Assembly Transcript 1) [19]. Suppression of NEAT1 in LPS-treated mesangial cells reduced NF-κB activation via miR-204 increase. Similar results were obtained in LPS-treated HK-2 cells, demonstrating that NEAT1 deficiency protected against LPS injury by targeting miR-22-3p/NF-κB axis [20] and let-7b-5p/TRAF6 axis [21]. Furthermore, NEAT1 downregulation promoted M2-macrophage polarization via miR-125a-5p/TRAF6/TAK1 axis and ameliorated LPS-harmful effects in RAW264.7 cells [22]. The implication of NEAT1 in SI-AKI has also been observed *in vivo*. NEAT1 silencing reduced kidney injury, ameliorated renal function, inflammation (TNF-α, IL-1β, IL-6), lipid peroxidation and cell death via miR-27a-3p/TAB3 axis in a cecal ligation puncture induced SI-AKI model in rats [23].

The lncRNA taurine upregulated gene 1 (TUG1) has also been studied in SI-AKI. Overexpression of TUG1 was protective in HK-2 cells treated with LPS by inhibiting miR-223 expression and modulating NF-kB [24], whereas TUG1 suppression increased LPS-mediated injury [25]. Interestingly, TUG1 protective effects against LPS-injury depends on the interaction with the transcription factor Nrf2 (Nuclear factor erythroid 2-related factor 2) [26]. HOXA cluster antisense RNA 2 (HOXA-AS2) is another lncRNA involved in SI-AKI. HOXA-AS2 expression is decreased in septic patients, SI-AKI experimental models and LPS-treated HK-2 cells [27]. In this study, overexpression of HOXA-AS2 played a protective role in this syndrome via miR-106b-5p and inhibiting the Wnt/β-catenin and NF-κB pathways. The lncRNA, CASC2 plasma levels were reduced and negatively correlated with AKI severity in septic patients [24]. In vitro experiments in HK-2 cells reported the protective role of CASC2 against LPS-mediated apoptosis, oxidative stress and inflammation by suppressing the miR-155 and NF-κB signaling pathway [28].

The lncRNA TapSAKI levels have been found to be elevated in urine from SI-AKI rats [29]. TapSAKI induced cell death and inflammatory response via miR-22/PTEN/TLR4/NF-κB in LPS-treated HK-2 tubular cells. In another study, overexpression of lncRNA PVT1 in HK-2 treated with LPS and PVT1 downregulation increased the cell growth and reduced inflammation by inhibiting JNK/NF-κB and TNFα signaling pathways [30]. Upregulation of the lncRNA HOX transcript antisense RNA (HOTAIR) was observed in LPS-cultured HK-2 cells and in an SI-AKI experimental model induced by injection of *Escherichia coli* suspension into the distal ureter of rats [31]. HOTAIR inhibited miR-22 and promoted cell death and HMGB1 expression, whereas its suppression ameliorated renal function. In contrast with this data, a later study in SI-AKI induced by cecal ligation puncture demonstrated that HOTAIR overexpression reduced renal damage, inflammation and cell death in the kidney by downregulating the miR-34a/Bcl-2 signaling pathway [32]. Recently, inhibition of the lncRNA MEG3 (maternally expressed gene 3) reduced LPS-mediated cell death via miR-21/PDCD4 in vitro in kidney proximal tubular epithelial-like cells [33].

A recent study identified DANCR as a lncRNA decreased in blood serum samples of patients with SI-AKI and in HK-2 cells treated with LPS [34]. DANCR promoted cell viability and inhibited cell death by sequestering miR-214 and regulating Krüppel-like factor 6 expression. LncRNA CRNDE was also downregulated in an experimental model of urine-derived SI-AKI and in HK-2 and HEK293 cells treated with LPS [35]. CRNDE deficiency decreased renal function and increased cell death by inhibiting miR-181a-5p/PPARα pathway, while CRNDE overexpression proved beneficial against LPS-mediated effects. Opposite results were reported in another experimental model of SI-AKI by LPS injection. CRNDE was upregulated in LPS injected mice and its inhibition improved renal function by regulating the TLR3/NF-κB pathway [36]. Similarly, another study demonstrated that CRNDE overexpression activated TLR4/NF-κB pathway via miR-146a, increasing inflammation and cell death in HK-2 cells treated with LPS [37].

The lncRNA SIKIAT1 (sepsis-induced kidney injury associated transcript 1) plays a key role in SI-AKI by increasing cell death in HK-2 cells treated with LPS via miR-96/FOXA1 axis [38]. Another lncRNA involved in SI-AKI is MIAT (myocardial infarction associated transcript). MIAT was upregulated in kidneys from LPS-injected rats and in NRK-52E cells treated with LPS, while miR-29a was decreased, suggesting an interaction between MIAT and miR-29a [39]. Moreover, in the lncRNA Whey acidic protein/Four-Disulfide Core domain 21 (Wfdc21), renal expression was increased in a cecal ligation and puncture SI-AKI model in mice and in LPS-stimulated RAW264.7 cells [40]. Wfdc21 downregulation modulated inflammatory Stat3 and TLR4 signaling pathways.

The expression of the lncRNA LINC00261 was significantly decreased in the serum of patients with sepsis and in a murine model of LPS injection [41]. In HK-2 cells treated with LPS, LINC00261 overexpression inhibited cell death and inflammation via the miR-654-5p/SOCS3/NF-κB pathway. A recent study described the lncRNA DLX6 antisense RNA 1 (DLX6-AS1) to be increased in serum of septic AKI patients and in HK-2 cells treated with LPS [42]. This study concluded that DLX6-AS1 was involved in LPS-mediated pyroptosis via miR-223-3p/NLRP3 axis in vitro. The lncRNA small nuclear RNA host gene 5 (SNHG5) was also increased in the serum of patients with SI-AKI [43]. SNHG5 suppression inhibited cell death and inflammation response via miR-374a-3p/TLR4/NF-κB pathway. As in the case of SNHG5, the lncRNA small nucleolus RNA host gene 14 (SNHG14) was increased in LPS-treated HK-2 cells due to the activation of the TLR4/NF-κB pathway. This study demonstrated that SNHG14 plays a key role in cellular injury due to LPS by increasing oxidative stress, inflammation and cell death through activation of IRAK4/NF-κB and IL-6R/STAT3 signaling via miR-93 [44]. Finally, the role of lncRNAs has also been studied in LPS-induced podocyte damage. The lncRNA GAS5 expression decreased in a time-dependent manner in LPS-stimulated podocytes. Following GAS5 knockdown by siRNA, nephrin expression and function of podocyte barrier was decreased by reducing PTEN expression via PI3K/AKT [45].

#### 2.1.2. Ischemia-Reperfusion

One of the most common processes inducing AKI is ischemia reperfusion (I/R). I/R injury is caused by the blockade of blood flow, leading to hypoxia and accumulation of metabolic products in the kidney, followed by a period of reperfusion that enhances renal injury by activating an inflammatory response [9,11,46].

A recent RNAseq study determined the lncRNA expression profile in a murine model of I/R, showing 5868 lncRNAs differentially expressed in the kidney [47]. In this study, the lncRNA NONHSAT183385.1 was increased in human tubular epithelial cells after hypoxia and reoxygenation, suggesting a key role in this pathological setting. 2267 differentially expressed lncRNAs were also observed in experimental I/R [48]. Microarray data were further confirmed by gene expression analysis, showing upregulation of several lncRNAs (ENSMUST00000124572, ENSMUST00000097928, ENSMUST00000180989, ENSMUST00000147219, uc007mos1 and ENSMUST00000169128) and downregulation of NR_040589, ENSMUST00000145410, ENSMUST00000139773, and AK078749. Gene ontology analysis showed that these lncRNAs participated in multiple biological processes in I/R, including glycine, serine, threonine metabolism and inflammatory pathways.

The lncRNA X-inactive specific transcript (*Xist*) has been shown to increase in I/R damaged kidneys and in renal cells after hypoxia/reoxygenation treatment, inducing apoptosis and inflammation through the modulation of miR-124-3p and its target gene ITGB1 [49]. Other authors have confirmed upregulation of Xist in I/R and identified the role of this lncRNA in the modulation of miR142-5p and PDCD4 [50]. The lncRNAs H19, MALAT1 and NEAT1 were also upregulated in human biopsies of AKI, experimental models of I/R and in cultured hypoxic endothelial and tubular cells [51,52,53,54]. Overexpression of the lncRNA H19 in an experimental I/R model, improved renal function and angiogenesis, and diminished inflammation and apoptosis throughout regulation of miR-30a-5p [51]. However, gene deletion of MALAT1 in mice did not improve renal function, tubular injury, inflammation or fibrosis [52]. In another study, downregulation of MALAT1 in cultured hypoxic cells significantly increased expression of HIF-1α, IL-6, TNF-α and NF-κB [53]. In vitro gene blockade of NEAT1 reduced hypoxia-mediated apoptosis (Bax/Bcl-2 ratio) and increased miR27a-3p expression, identifying this miRNA as a target of NEAT1 [54].

In cultured tubular cells, upregulation of the lncRNA EGOT diminished hypoxia/reoxygenation mediated autophagy via interaction with the RNA-binding protein Hu antigen R (HuR) and further regulation of the ATG7/16L1 expressions [55]. LncRNA GAS5 was induced by I/R injury, promoting renal apoptosis by regulation of the miR-21/TSP-1 axis [55]. The lncRNA LINC00520 was upregulated in I/R injured rats and in hypoxic tubular epithelial cells, being associated with a reduction of miR-27b and increased levels of the Oncostatin M receptor (OSMR) [56]. The gene blockade of LINC00520 reduced renal damage through the attenuation of OSMR levels and PI3K/AKT signaling pathway [56]. In addition, the gene blockade of LncRNA SNHG14 in I/R reduced inflammation and oxidative stress by modulating miR-124-3p, which regulates MMP2 levels [57]. In two recent studies, HK2 cells under hypoxia increased HIF-1α levels through the LncRNA PRINS and its interaction with CCL5 [58]. The LncRNA MEG3 was also found to be elevated in human tubuloepithelial cells, aggravating hypoxia/reoxygenation induced apoptosis throughout regulation of the miR-129-5p/HMGB1 axis [59]. All these results confirm the emerging role of LncRNAs in the pathophysiology of I/R renal injury and their possible use as biomarkers in this pathological setting.

**Table 1 ijms-22-06077-t001:** Effects of LncRNAs in experimental AKI.

AKI Model	LncRNA	Targeted miRNA	miRNA Beneficial Effects	References
Sepsis	lncRNA 6406	miR-687	Reduces inflammation, oxidative stress and cell death.	[13]
	TCONS_00016233	miR-22-3p	Inhibits apoptosis and TLR4-mediated inflammatory response.	[15]
	MALAT1	miR-146a	Reduces cell death and NF-κB mediated inflammation.	[16,17,18]
	NEAT1	miR-204	Decreases oxidative stress, apoptosis and NF-κB associated inflammation.	[19]
		miR-22-3p	Reduces oxidative stress, cell death, autophagy and inflammation	[20]
		miR-125a-5p	Decreases inflammation by promoting macrophage M2 polarization.	[22]
		miR-27a-3p	Alleviates oxidative stress, apoptosis and reduces kidney injury.	[23]
	TUG1	miR-223	Reduces oxidative stress and cell death by activating Nrf2 pathway.	[24,25,26]
	HOXA-AS2	miR-106b-5p	Inhibits NF-κB/Wnt/β-catenin pathways and reduces renal damage.	[27]
	CASC2	miR-155	Suppresses NF- κB mediated inflammatory cytokines expression.	[28]
	TapSAKI	miR-22	Inhibits cell apoptosis and inflammation.	[29,30]
	HOTAIR	miR-22	Decreases cell death and improves renal function.	[31]
		miR-34a	Improves renal function, reduces cell death and inflammation.	[32]
	MEG3	miR-21	Reduces cell apoptosis.	[33]
	DANCR	miR-214	Suppresses cell death and inflammatory cytokines production.	[34]
	CRNDE	miR-181a-5p	Induces cell proliferation and decreases cell death and inflammation.	[35]
		miR-146a	Reduces cell death and inflammatory TLR4/NF-κB signaling pathway.	[37]
	SIKIAT1	miR-96	Decreases cell apoptosis.	[38]
	MIAT	miR-29a	Reduces cell death by inhibiting caspase 8.	[39]
	LINC00261	miR-654-5p	Suppresses cell apoptosis and inflammation via. NF-κB inhibition.	[41]
	DLX6-AS1	miR-223-3p	Decreases NLRP3 expression and pyroptosis.	[42]
	SNHG5	miR-374a-3p	Reduces NF-κB/TLR4 mediated inflammation and cell death.	[43]
	SNHG14	miR-93	Ameliorates oxidative stress, cell death and inflammation by activating IRAK4/NF-κB signaling.	[44]
I/R	XIST	miR-124-3p	Improves renal function and decreases inflammation.	[49]
		miR142-5p	Reduces renal injury and inflammation.	[50]
	H19	miR-30a-5p	Decreases cell death and inflammation.	[51]
	MALAT1	miR-146a	Improves renal injury, inflammation, and fibrosis.	[52,53]
	NEAT1	miR27a-3p	Ameliorates hypoxia-mediated cell death.	[54]
	GAS5	miR-21	Reduces renal injury and cell death.	[60]
	LINC00520	miR-27b-3p	Improves renal injury and cell death.	[56]
	SNHG14	miR-124-3p	Decreases inflammation and oxidative stress.	[57]
	MEG3	miR-129-5p	Reduces tubuloepithelial cell death.	[59]
Cisplatin	XLOC-032768		Decreases apoptosis and inflammation.	[61]
	LRNA9884		Reduces tubular cell death and inflammation.	[62]
UUO	TCONS_00088786		Reduces TGF-β/Smad mediated fibrosis.	[63]
	TCONS_01496394			[63]
	Gm10251		Decreases fibrosis in tubular cells.	[64]
	Fam120aos			[64]
	Gm16076			[64]
	lncRNA 74.1		Reduces fibrosis by inhibiting Nrf2-Keap1 pathway.	[65,66]
	MIAT		Targets α-SMAD and reduces fibrosis.	[67,68]
	H19			[69]
	HOTAIR	miR-124	Reduces fibrosis by inhibiting Notch1.	[70,71]

#### 2.1.3. Nephrotoxic Agents

One of the most remarkable nephrotoxic agents is cisplatin, which induces inflammation, cell death and tubular cell injury [72,73]. Zhou et al. explored the role of lncRNA XLOC-032768 in cisplatin-induced AKI in vivo and in vitro. Their results show that overexpression of this lncRNA decreased apoptosis and TNF-mediated inflammation in mice and cells exposed to cisplatin [61]. The lncRNA LRNA9884 was also increased in the nucleus of renal epithelial tubular cells in a cisplatin-induced AKI model [62].

#### 2.1.4. Obstructive AKI

Obstructive nephropathy is a recurrent cause of AKI [74]. If not treated in time, AKI associated to obstructive nephropathy can eventually lead to chronic kidney disease (CKD). The best experimental approach to study the AKI–CKD transition is the murine model of unilateral ureteral obstruction (UUO) [75]. The UUO model is characterized by a reduced GFR and blood flow, oxidative stress, renal cell death and tubulointerstitial fibrosis. All these phenomena contribute to a progressive loss of renal function [76,77].

Sun et al. examined the differential expression of lncRNAs in kidneys from UUO mice. They observed that 24 lncRNAs were upregulated and 79 lncRNAs downregulated [63]. Specifically, TGF-β regulated the expression of lncRNAs TCONS_00088786 and TCONS_01496394, which are involved in the modulation of the profibrotic TGF-β/Smad pathway. Additional urine analysis reported upregulation of 625 lncRNAs and downregulation of 177 lncRNAs, some of which (NONRATT044682, 361619 and 689064) could be possible biomarkers of renal fibrosis. In another study, lncRNAs Gm10251, Fam120aos and Gm16076 regulated the Jak/STAT pathway, whereas lncRNAs Fam12aos and Gm16076 interacted with fibroblast integrin beta1 (ITGB1) [64]. The lncRNA 74.1 was downregulated in HK-2 cells exposed to TGF-β. Moreover, lcRNA 74.1 overexpression reduced fibrosis by activating autophagy and the Nrf2-keap1 pathway in cultured HK-2 cells and the UUO model [65,66].

TGF-β upregulated lncRNAs MIAT and H19 in renal cells, suggesting an important role of these lncRNA in renal interstitial fibrosis [67,68,69]. Supporting this hypothesis, deletion of H19 reduced production of α-SMA and collagen IV in UUO mice and HK-2 cells [69].

lncRNA HOTAIR expression was upregulated in UUO, inhibiting miR-124 expression in UUO mice [71]. Since miR-124 blocked the Notch1 signaling pathway and prevented fibrosis and epithelial to mesenchymal transition, HOTAIR suppression promoted the development of fibrosis and chronic renal failure [70]. Finally, deficiency of the lncRNAGas5 aggravated renal fibrosis in the UUO mice [78].

### 2.2. Chronic Kidney Disease

CKD is defined by an estimated glomerular filtration rate (eGFR) of less than 60 mL/min/1.73 m^2^. Hypertension and heart disease constitute the majority of cardiovascular (CV) risk factors found in CKD patients [79] and are the primary cause of death due to complications, such as anemia [80]. Several miRNAs have been associated with CKD and CV diseases. In a study by Fournidier et al., the authors evaluated the role of two circulating miRNAs (miR-223 and miR-126) in this pathology [81]. They reported that decreased miR-126 and miR-223 serum levels were associated with mortality, CV events and lower eGFR at different CKD stages. They observed that the link between these two miRNAs serum levels, the mortality, the CV diseases and the renal events appears to depend on the eGFR in CKD. Non-coding RNAs could be, thus, useful as diagnostic or prognostic biomarkers in CKD patients. Unfortunately, multiple issues limit the use of miRNAs as therapeutic targets for CKD, such as ubiquitous tissular expression, toxicity or off-target effects [82]. To date, most of the research on ncRNAs and CKD has been focused on miRNAs but not much is known about the role of lncRNAs. In 2016, the study of Khurana et al. identified urinary exosomal ncRNAs (exo-ncRNAs) as new biomarkers for early CKD diagnosis [83]. By RNASeq and novel computational algorithm analysis, the authors identified 30 differentially expressed ncRNA (including 16 miRNAs, 2 tRFs, 3 mt-tRNAs and 9 lincRNAs) in the early stages of CKD compared to healthy controls. Among these, 16 miRNAs showed significant alterations in their abundance. Nine were significantly increased and seven were significantly decreased in early CKD stages compared to healthy controls. Interestingly, miRNA-181a is the one whose exosomal abundance was significantly decreased by about 200-fold in all four stages of CKD in patients compared to healthy controls. Thus, in the future, miR-181a could be employed as a potential biomarker for early detection of CKD. In addition to miRNAs, the study has identified in CKD patients, nine antisense RNAs differentially present in exosomes compared to healthy controls. They also observed that two tRFs were significantly decreased in exosomes of CKD patients vs. healthy controls. Three exosomal mt-tRNAs were also significantly decreased in CKD patients as compared to healthy controls. Furthermore, this study investigated whether urinary extracellular RNAs (exRNAs) are derived from kidney related cells. For this matter, they employed a cell culture system reflecting CKD by exposing renal proximal tubular epithelial cells (RPTECs) to oncostatin M (OSM), TGF-β1 and IL-1β. They found some evidence that urinary exRNAs might originate from kidney-related cell types. Finally, this study identified a significant number of potential diagnostic biomarkers that might be employed in CKD in future research.

Another study published in 2019 found that lncRNAs NOP14-AS1 and HCP5 were potential prognostic biomarkers in CKD [84]. They first compared the expression profiles of several lncRNAs in healthy individuals and CKD patients with normal controls. They screened 821 significantly differentially expressed mRNAs and lncRNAs using Limma (www.bioconductor.org/packages/release/bioc/html/limma.html (accessed on 22 January 2021) between CKD and control samples. A lncRNA-miRNA-mRNA-pathway network using these overlapping pathways, revealed that lncRNAs of HCP5 and NOP14-AS1 and genes of CCND2, COL3A1, COL4A1 and RAC2 were significantly correlated with CKD. In summary, this study suggested that several lncRNAs (i.e., NOP14-AS1 and HCP5) were potential prognostic biomarkers for CKD progression.

Furthermore, Chun-Fu Lai et al. reported that elevated plasma level of lncRNA DKFZP434I0714 was a reliable biomarker in uremic patients to predict adverse CV outcomes [85]. The authors validated that elevated plasma level of lncRNA DKFZP434I0714 predicts adverse CV outcomes and death in patients with end stage renal disease (ESRD). Further analyses revealed that lncRNA DKFZP434I0714 was increased in human aortic endothelial cells (HAEC) stressed with hypoxia and is involved in endothelial dysfunction and inflammation. Thus, knocking down DKFZP434I0714 inhibited the expression of ICAM-1 (intracellular adhesion molecule 1) and VCAM-1 (vascular cells adhesion molecule 1), increased the expression of eNOS (endothelial nitric oxide synthase) and reduced hypoxia-mediated endothelial cell apoptosis and monocyte adhesion.

Taken together, these data collectively suggest a potential role of lncRNA DKFZP434I0714 as a new class of prognostic biomarker of adverse CV outcomes in ESRD patients, showing its involvement in the pathogenesis of endothelial dysfunction, vascular inflammation and atherosclerosis. Indeed, the present study revealed that lncRNA DKFZP434I0714 is involved in the pathogenesis of endothelial dysfunction, vascular inflammation and atherosclerosis.

Santer et al. studied whether the presence of CKD modified the association of plasma LIPCAR (long intergenic non-coding RNA predicting cardiac remodeling) with left ventricular (LV) remodeling and CV outcomes in patients with heart failure (HF) [86]. In this study, plasma LIPCAR levels were independently associated with higher risk of hospitalization in elderly HF patients without CKD. Plasma LIPCAR could be a predictor of HF outcomes in elderly patients without CKD. Table 2 summarizes some of the ncRN prognostic biomarkers for CKD.

Arbiol-Roca et al. were the first to study the association of antisense RNA in the INK4 locus (ANRIL) polymorphisms with Major Adverse CV Events (MACE) in CKD patients in Hemodialysis (HD) [87]. HD is a therapeutic procedure to remove fluid and waste products from the patient’s blood used in the management of acute and chronic renal disease. Using a multivariate model, ANRIL polymorphism rs10757278 was the only one that showed a statistically significant relationship with MACE and diabetes mellitus. Those findings of ANRIL polymorphisms may contribute in the future to the management of MACE in the HD population.

A recent study in human aortic smooth muscle cells (HA-VSMCs) by Bao et al. found that osteogenic differentiation induced by high phosphorus may be regulated by eight lncRNAs (NONHSAT058810.2, NONHSAT197162.1, NONHSAT 033640.2, NONHSAT036152.2, NONHSAT179247.1, NONHSAT162315.1, NONHSAT061050.2 and NONHSAT006046.2) [88]. All these lncRNAs may regulate the expression of transcription factors (TFs), such as STAT1, KAT2A, GATA2, TAF7 and REST. Altogether, this working list of lncRNAs may be associated with osteogenic differentiation induced by high phosphorus.

Another recent study aimed to understand the transcriptional profile of the proximal tubule (PT) during kidney fibrosis to explain why CKD progresses faster in males versus females [89]. They used the UUO mouse model of kidney fibrosis in which they performed translating ribosome affinity purification (TRAP). The study identified a total of 439 lncRNAs expressed in PT, of which, 143 underwent differential regulation during fibrosis. RNA in situ hybridization (ISH) of Snhg18 (high expression) and Gm20513 (low expression), two representative lncRNAs, confirmed the fibrotic-induced expression patterns. Subsequently, they focused on the potential target mRNAs that could be regulated by Snhg18 and Gm20513. Interestingly, the putative target genes were related to proinflammatory genes such as Fcgr2b, Cx3cr1, Lpl and Alox5ap. Finally, they implemented a data-driven approach to identify key transcription factors that lead to disease progression in kidney fibrosis. They identified proinflammatory and profibrotic transcription factors such as Irf1, Nf-kb1 and Stat3 as potential drivers of fibrosis regulation.

LncRNAs have also been associated with renal fibrosis via TGF-β, a master regulator of fibrosis that promotes renal fibrosis, via lncRNAs implicated in TGF-β1 isoform [90]. Although TGF-β1 has a dominant role to promote renal fibrosis, its effector Smad proteins (Smad2, Smad3 and Smad4) exert distinct and even opposing functions in the regulation of fibrosis. Smad3 exerts a profibrotic function and can induce transcription of profibrotic miRNAs and lncRNAs [91]. RNA sequencing and array analysis have identified approximately 21 lncRNAs induced by TGF-β/Smad3 signaling in the kidney. Among them, the lncRNAs np_5318 and np_17856 were involved in TGF-β-mediated renal fibrosis [92]. Other studies identified lncRNA np_4334 (also named Ptprd-IR) as a novel molecular target for TGF-β1-mediated nephritis [93]. They found that Ptprd-IR knockdown attenuated NF-κB activation and its inflammatory targets in cultured tubular epithelial cells and UUO mice. They also discovered a highly conserved site for Smad3 binding within the Ptprd-IR promoter. These findings open novel therapies against inflammation and fibrosis in CKD by targeting lncRNAs, although data are currently limited.

### 2.3. Diabetic Nephropathy

In the publication of Long et al., LncRNA taurine-upregulated 1 (Tug1) has been reported to play a role in diabetic nephropathy (DN) [25,94]. In these studies, authors performed RNA-seq to identify changes in lncRNA levels. They found that glomerular Tug1 levels were decreased in diabetic mice as well as in renal biopsies from diabetic patients. The authors demonstrated that Tug1 regulates mitochondrial function in podocytes via PGC-1α (Peroxisome proliferator activated receptor-γ Coactivator-1 α). Moreover, transgenic mice that overexpressed Tug1 specifically in podocytes were protected from diabetes-induced CKD, suggesting that this lncRNA may be a possible therapeutic target to treat kidney disease and/or diabetes.

RNA deep-sequencing techniques in glomeruli from streptozotocin-induced diabetes identified a cluster of nearly 40 miRNAs regulated by the lncRNA lnc-Megacluster (lnc-MGC) [95]. This lncRNA was under the control of CHOP (endoplasmic reticulum stress-related transcription factor). Inhibition of lnc-MGC decreased the expression of key cluster kidney miRNAs, triggering early diabetic nephropathy (DN).

Xist has already shown to be active in several roles, including X-chromosome silencing and tumoral progression. Xist was recently shown to be deregulated in DN [96]. Xist was overexpressed in experimental DN and renal biopsies from DN patients. Moreover, Xist downregulation triggered miR-93-5p overexpression, which in turn diminished Cyclin Dependent Kinase Inhibitor 1A (CDKN1A) expression, alleviating renal interstitial fibrosis in DN. This work demonstrated that silencing Xist may provide a future strategy to stop the progression of DN [96]. A close relationship between lncRNA KCNQ1 opposite strand/antisense transcript 1 (KCNQ1OT1)/miR-18b-5p/Sorbin and SH3 Domain Containing 2 (SORBS2) was also described in DN [97]. The authors showed that either silencing of SORBS2 or knockdown of KCNQ1OT1 diminished NF-ĸB-mediated proliferation, fibrosis and apoptosis in renal cells exposed to high glucose concentrations.

Several miRNAs have been identified as possible biomarkers for diabetes and DN, but few researchers have addressed the involvement of lncRNAs in this pathological context. Yang et al. studied the lncRNA profiles of patients with diabetes mellitus and DN [47]. In this article, 245 lncRNAs were upregulated and 680 downregulated in the serum of diabetic patients as compared with healthy individuals, while 45 and 813 lncRNAs were up- and downregulated, respectively, in the serum of DN patients compared with diabetic patients [47] Another interesting diabetes-related mechanism was found concerning angiotensin II type 1 receptor-associated protein (ARAP). ARAP is involved in the trafficking of other receptors, including EGFR and TRAIL-R that may also contribute to deleterious effects of diabetes in the kidney. lncRNA-ARAP1-AS1 and ARAP1-AS2 enhance ARAP1 mRNA expression and may be involved in the pathogenesis of diabetes and DN. Indeed, circulating lncRNA-ARAP1-AS1, ARAP1-AS2 and ARAP1 may serve as new biomarkers for diabetes and DN. ARAP1 regulates blood pressure and renal physiology by activating the intrarenal renin–angiotensin system [98]. Inhibiting proteins of the ARAP1/AT1R signaling pathway decreases kidney damage by alleviating renal inflammation, extracellular matrix accumulation and fibrosis [99]. It is, thus, probable that lncRNAs regulating the ARAP1 transcript are implicated in the course of DN. Conversely, Li et al. published the role of ARAP1 and its antisense RNA, ARAP1-AS2, in the pathogenesis of diabetes in cytoskeleton rearrangement under high glucose conditions and epithelial–mesenchymal transition indicating an important role in diabetic renal fibrosis [100].

### 2.4. Glomerular Diseases

#### 2.4.1. Lupus Nephritis

Increased expression of the pro-apoptotic lncRNA-p21 was found in monocytes and urinary cells from patients with lupus nephritis (LN), correlating with the activity of disease (33,396,699). In this article, the authors found that LN mice showed a progressive increase in lncRNA-p21. The lncRNAs RP11-2B6.2 and CircHLA-C were increased in renal biopsies from patients with LN [101,102]. RP11-2B6.2 was positively correlated with the inflammatory status of the disease [101], whereas CircHLA-C correlated with proteinuria, creatinine and crescentic glomeruli [102]. Experimental studies in renal cells demonstrated that the lncRNA RP11-2B6.2 inhibited the suppressor of cytokine signaling 1 (SOCS1), thus reducing type 1 interferon [101]. LncRNA found in dendritic cells (lnc-DC) was also found to be elevated in patients with LN, mainly in those with active LN [103]. As well, circRNA_002453 plasma level was found to be elevated in patients with LN, being positively correlated with proteinuria and kidney disease activity [104].

#### 2.4.2. Focal Segmental Glomerulosclerosis

Patients with focal segmental glomerulosclerosis (FSGS) showed higher expression of LncRNA LOC105375913 [105]. Further experiments in tubular cells and experimental mice demonstrated that LOC105375913 inhibited miR-27b, thus promoting overexpression of Snail and tubulointerstitial fibrosis. This LncRNA was activated by the C3a/p38/XBP-1s pathway [105]. In other study, it was demonstrated that LncRNA LOC105374325 caused podocyte injury in individuals with FSGS [106]. The authors showed that LOC105374325 reduced miR-34c and miR-196a/b levels, resulting in an increased expression of the pro-apoptotic proteins Bax and Bak. LOC105374325 upregulation in this pathological setting may be related to activation of the P38/C/EBPβ pathway in the podocytes of these individuals.

#### 2.4.3. IgA Nephropathy

Many lncRNAs are differently expressed in patients with IgA-negative Mesangial proliferative glomerulonephritis and may contribute to disease development [107]. More recently, a microarray analysis in monocytes from IgA nephropathy patients and healthy individuals identified more than 250 differentially expressed lncRNAs and mRNAs, mainly involved in the regulation of innate immune response [108]. Similar results were found in other studies applying a system biology approach, where 217 lncRNA differentially expressed in PBMCs were suggested as potential factors involved in IgA nephropathy pathophysiology [109]. In this article, HOTAIR was the topmost lncRNA in regulating differentially expressed genes/miRNAs in IgA nephropathy. A recent study determined serum levels of exosomal lncRNA in patients with IgAN and found that the lncRNA-G21551 was down-regulated and may be a potential surrogate biomarker of the disease [110].

#### 2.4.4. Membranous Nephropathy

The role of lncRNA has also been explored in membranous nephropathy (MN), where increased levels of the lncRNA Xist were found in urinary samples of MN patients and kidneys from mice with MN [111]. In cultured podocytes, down-regulation of lncRNA Xist inhibited angiotensin II-mediated apoptosis by suppressing the miR-217-TLR4 pathway [112].

### 2.5. Renal Cancers

As with proteins and miRNAs, lncRNAs can be divided into two categories cancer-wise: the pro-tumoral (increased in the tumour) and the tumour suppressors (decreased in the tumour). Unsurprisingly when dealing with gene regulators, many lncRNA transcripts are associated with tumoral progression such as *Xist, one of the best characterized lncRNAs* [2]. *Xist* is located on the long (q) arm of the X chromosome in placental mammals and is instrumental in the X-inactivation process enabling equal expression of the X and autosomal chromosomes. Xist maintains dosage compensation for approximately 1000 genes on the X chromosome, some of which are possible oncogenes. Xist expression has been shown to be decreased in renal cell carcinoma [113]. Additionally, the authors showed that Xist over-expression decreases proliferation and is a stop signal at stage G0/G1 by targeting the miR-106b/P21 axis. The lncRNA ANRIL is overexpressed in several cancers, including malignant breast cells, and its subcellular location is an indicator of cancer progression [114]. In general, *ANRIL* overexpression in tumour cells favours proliferation and cell survival, while *its inhibition* decreases tumour mass and increases apoptosis. The same is true in kidneys, as ANRIL overexpression favours the invasive potential of renal carcinoma cells by impacting the β-catenin pathway [115]. The role of MALAT1 RNA, a highly expressed and conserved transcript, in carcinogenesis, has been abundantly documented [116]. In renal cell carcinoma, MALAT1 is highly expressed and interacts with miR-203 and BIRC5 to increase cell proliferation [117]. These are some of the most striking examples of lncRNAs involved in renal cancer, for which there is a comprehensive review [118].

## 3. Perspectives in Therapy

LncRNAs are prime candidates as biomarkers in the nephrological field (Figure 1). Several studies, summarized in this review, have shown that screening pre-clinical or cellular models can help to identify new lncRNAs that will be later developed as diagnostic and/or prognostic biomarkers. Compared to miRNAs, they have the advantage of being more tissue-specific. At this time, the main biofluid studied is the blood (as summarized in this review) but urinary expression of lncRNAs also has a strong biomarker potential. However, to the best of our knowledge, few studies have been published in the nephrology field. The study of Khurana et al., that identified urinary exosomal ncRNAs in CKD patients, was already discussed in the CKD sub-chapter [83]. A recent study in the related urological field shows that a urinal lncRNA panel has potential clinical value to diagnose bladder cancer [119]. Their smaller counterparts, miRNA, were also recently shown to have the same potential in the same organ, but the authors noted the importance of the controls being age- and sex-related [120]; this also proves true when measuring lncRNAs.

LncRNAs are closely related to progression of renal damage and therapeutic strategies targeting these molecules may be useful to ameliorate this pathology [121]. The development of new therapies using RNA interference is a present challenge, and miRNAs have an important role in the regulation of several kidney pathologies. Several carriers such as exosomes, microvesicles and high-density lipoproteins have been shown to carry functional miRNAs in association with other compounds (lipids, proteins and mRNAs). (Figure 2) These naturally occurring nanomaterials can be used to deliver functional lncRNAs to specific renal cells [122]. Extrinsic nanotechnology vehicles can also be used, such as 13 nm wide gold nanoparticles functionalized with monolayers of alkylthiol-modified RNA molecules. CRISPR/cas9 system is another system useful in the non-coding RNA field, as specific CRISPR/cas9 scissors can be used to target selected sites to abolish the long-term expression of RNAs [6]. Thus, lncRNAs, miRNAs or antisense sequences can be delivered to the kidney (depending on the context).

## 4. Conclusions

lncRNAs have been identified as key molecules involved in the pathophysiology of renal injury and may be useful as potential biomarkers for the early diagnosis and prognosis of patients with kidney diseases (Figure 1). New biomarkers are sorely needed in the field of renal diseases in order to have a better prognosis, and it is our belief that lncRNAs can be welcome candidates. Furthermore, regulation of these non-coding RNAs opens new therapeutic approaches to decrease renal damage, although it is necessary to develop new studies to fully address this issue. New technological tools have appeared, such as nanotechnologies and CrispR/Cas9 (Figure 2), and we can be sure that they will facilitate the transition from the bench to the clinical bed.

## Figures and Tables

**Figure 1 ijms-22-06077-f001:**
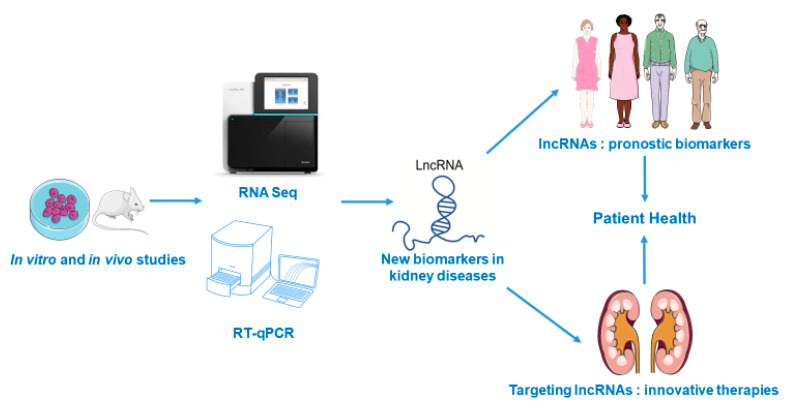
Use of lncRNAs as new prognostic biomarkers and novel therapeutic targets in renal diseases.

**Figure 2 ijms-22-06077-f002:**
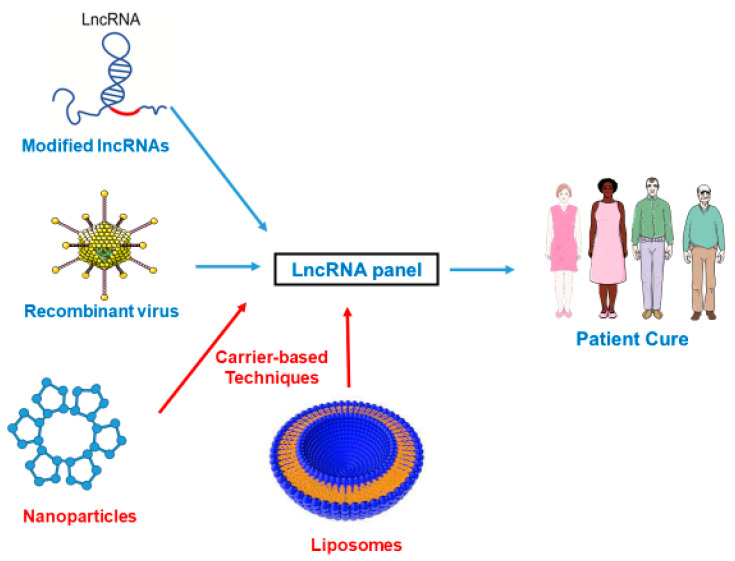
Innovative therapies targeting lncRNAs in the kidney.

**Table 2 ijms-22-06077-t002:** Representative ncRNAs as prognostic biomarkers.

ncRNAs	Prediction/Biomarker	Reference
Urinary exosomal ncRNAs -miRNA-181a -9 antisense RNAs (EAF1-AS1, PCBP1-AS1, RP11-178F10.1, RP11-315I20.1, RP11-378E13.4, RP11-68I3.2, RP11-700F16.3, RP11-98D18.1 and RP11-1382.1) -trf^Val^, trf^Leu^	Diagnostic biomarker for early detection of CKD	[83]
LncRNA NOP14-AS1 and LncRNA HCP5	Predicting CKD progression	[84]
LncRNA DKFZP434I0714	Predicting adverse CV outcomes in ESRD patients	[85]
LncRNA LIPCAR	Predicting HF outcomes in patients without CKD	[86]

Abbreviations: ncRNAs, non-coding RNAs; miRNA, microRNA; trf^Val^; trf^Leu^; CKD; LncRNA, long non-coding RNA; CV, Cardiovascular; ESRD, end-stage renal disease; LIPCAR, long intergenic non-coding RNA predicting cardiac remodeling; HF, Heart Failure.

## Data Availability

Not applicable.

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
