# Peer review of "Non-Coding RNAs in Kidney Diseases: The Long and Short of Them"

_ijms, 2021, doi:10.3390/ijms22116077_

Round 1

Reviewer 1 Report

Moreno and colleague have reviewed the existing literature on the role of non-coding RNAs (ncRNAs) in renal pathologies.   The authors comprehensively describe evidence for regulation of long ncRNAs in acute kidney injury, sepsis, ischemia, obstructive disease, chronic kidney disease and diabetes.   I have only minor issues with this well-written review.   

  1. (line 71). The authors discuss outlining the role of lncRNAs in cancer first, but never discuss this area in depth and this review concerns renal disease.  These statements regarding cancer should be deleted from review.  
  2. (line 156). Specify renal cell type for this lncRNA. 
  3. (line 160). Can the authors use a more appropriate term than sponging for interaction with miR-214.   Possibly sequester or bind to? 
  4. (line 281). Suggest replacing inferior to with less than. 
  5. (line 282). Can the authors define what cardiovascular disease is the primary cause of death in CKI as risk is not necessarily a cause of mortality?  
  6. (line 352). Authors need to define HD.
  7. (line 355). Sentence is incomplete – what does ANRIL do in the management of MACE? 
  8. (line 395). Delete “in”.
  9. (line 429). Add that “…ARAP1 facilitates expression of the AT1R to activate the intrarenal renin-angiotensin system”.
  10. (line 424). Authors should mention that ARAP is involved in trafficking of other receptors including EGFR and TRAIL-R that may also contribute to deleterious effects of diabetes in the kidney. 

Author Response

Reviewer1

Moreno and colleague have reviewed the existing literature on the role of non-coding RNAs (ncRNAs) in renal pathologies.   The authors comprehensively describe evidence for regulation of long ncRNAs in acute kidney injury, sepsis, ischemia, obstructive disease, chronic kidney disease and diabetes.   I have only minor issues with this well-written review.   

  1. (line 71). The authors discuss outlining the role of lncRNAs in cancer first, but never discuss this area in depth and this review concerns renal disease.  These statements regarding cancer should be deleted from review.  

Answer. We agree with the reviewer and the relevant paragraph has been deleted. Relevant references were removed.

  1. (line 156). Specify renal cell type for this lncRNA. 

Answer. We thank the reviewer and added the information «in kidney proximal tubular epithelial-like cells” line 156.

  1. (line 160). Can the authors use a more appropriate term than sponging for interaction with miR-214.   Possibly sequester or bind to? 

Answer. Actually sponge is a term widely used for this technical tool in the miRNA field. A sponge RNA contains multiple target sites complementary to a miRNA of interest, and is a dominant negative method. We thus propose to keep this term.

  1. (line 281). Suggest replacing inferior to with less than. 

Answer. We agree and this was done.

  1. (line 282). Can the authors define what cardiovascular disease is the primary cause of death in CKI as risk is not necessarily a cause of mortality?  

Answer. We agree with the reviewer and thank him for this remark which improved the manuscript. We added lines 282-284 the following sentence “. Hypertension and heart disease constitute the majority of cardiovascular (CV) risk factors found in CKD patients (78) and are the primary cause of death due to complications such as anemia (79).”. Two references were added and the ref list was changed accordingly.

  1. (line 352). Authors need to define HD.

Answer. We thank the reviewer for this query. HD was defined lines 353-355 “HD is a therapeutic procedure to remove fluid and waste products from the patient's blood used in the management of acute and chronic renal disease”

  1. (line 355). Sentence is incomplete – what does ANRIL do in the management of MACE? 

Answer. We clarified the sentence. It now reads “Those findings of ANRIL polymorphisms may contribute in the future in the management of MACE in HD population”

  1. (line 395). Delete “in”.

Answer. This has been done.

  1. (line 429). Add that “…ARAP1 facilitates expression of the AT1R to activate the intrarenal renin-angiotensin system”.

Answer. We thank the reviewer for this clever addition. Accordingly we added “ARAP1 facilitates expression of the AT1R to activate the intrarenal renin-angiotensin system” and a relevant reference was added.

  1. (line 424). Authors should mention that ARAP is involved in trafficking of other receptors including EGFR and TRAIL-R that may also contribute to deleterious effects of diabetes in the kidney. 

Answer. We thank the reviewer for this clever addition. Accordingly we added “ARAP is involved in trafficking of other receptors including EGFR and TRAIL-R that may also contribute to deleterious effects of diabetes in the kidney” lines 428-429.

Reviewer 2 Report

Moreno et al. summarized the role of lncRNA on kidney diseases and the molecular mechanisms involved. My comments are below.

Major:

  1. The role of lncRNA in normal renal physiology should be discussed.
  2. The role of lncRNA in glomerular diseases and renal cancers should be included in this MS.
  3. LncRNAs may be useful as potential biomarkers for the early diagnosis and prognosis of patients with kidney diseases. How about the relative importance of circulatory lncRNA vs. urinary lncRNA? This issue should be discussed.
  4. Is there any clinical evidences on the function of lncRNA in kidney diseases?

Mini:

   The references in table 1 are missing.

Author Response

Reviewer2

Moreno et al. summarized the role of lncRNA on kidney diseases and the molecular mechanisms involved. My comments are below.

Major

  1. The role of lncRNA in normal renal physiology should be discussed.

Answer. We agree with the reviewer and added a specific paragraph Page 2;

Physiologically, lncRNAs have been shown to modulate gene regulation at all levels (including but not limited to promoter activity, epigenetics, translation and transcription efficiency, intracellular trafficking …). (30103474) As is the case with other tissues, kidney expresses tens of thousands of lncRNAs sequences that are often conserved with coding genes. lncRNAs are divided into four categories: long intervening/intergenic noncoding RNAs, intronic lncRNAs, sense lncRNAs (including pseudogenes) and antisense lncRNAs. (30159410) The majority of lncRNAs are expressed in the nucleus, whereas a minority is cytoplasmic. (33898883) A promising feature of lncRNAs is that they are more tissue- or cell type-specific than mRNAs or miRNAs, suggesting a better biomarker potential.

  1. The role of lncRNA in glomerular diseases and renal cancers should be included in this MS.

Answer. We thank the reviewer for this insightful remark. We thus added two new paragraphs.

For glomerulonephritis, pages10-11 .

2.4 Glomerular diseases

2.4.1. Lupus nephritis

Increased expression of the pro-apoptotic lncRNA-p21 was found in monocytes and urinary cells from patients with lupus nephritis (LN), correlating with the activity of disease (33396699). In this article, the authors found that LN mice showed a progressive increase in lncRNA-p21. The lncRNAs RP11-2B6.2 and CircHLA-C were increased in renal biopsies from patients with LN (31130957, 29499937). RP11-2B6.2 was positively correlated with the inflammatory status of the disease (31130957), whereas CircHLA-C correlated with proteinuria, creatinine and crescentic glomeruli (29499937). Experimental studies in renal cells demonstrated that the lncRNA RP11-2B6.2 inhibited the suppressor of cytokine signaling 1 (SOCS1), thus reducing type 1 interferon (31130957). LncRNA found in dendritic cells (lnc-DC) was also found elevated in patients with LN, mainly in those with active LN (28423570). circRNA_002453 plasma level was also found elevated in patients with LN, being positively correlated with proteinuria and kidney disease activity (30172209)

2.4.2. Focal segmental glomerulosclerosis

Patients with focal segmental glomerulosclerosis (FSGS) showed higher expression of LncRNA LOC105375913 (30679767). Further experiments in tubular cells and experimental mice demonstrated that LOC105375913 inhibited miR-27b, thus promoting overexpression of Snail and tubulointerstitial fibrosis. This LncRNA was activated by the C3a/p38/XBP-1s pathway (30679767). In other study, it was demonstrated that LncRNA LOC105374325 caused podocyte injury in individuals with FSGS (30389788). The authors showed that LOC105374325 reduced miR-34c and miR-196a/b levels, resulting in increased expression of the pro-apoptotic proteins Bax and Bak. LOC105374325 upregulation in this pathological setting may be related with activation of the P38/C/EBPβ pathway in the podocytes of these individuals.

2.4.3. IgA Nephropathy

Many lncRNAs are differently expressed in patients with IgA-negative Mesangial proliferative glomerulonephritis and may contribute to disease development (22576627).  More recently, a microarray analysis in monocytes from IgA nephropathy patients and healthy individuals identified more than 250 differentially expressed lncRNAs and mRNAs, mainly involved in regulation of innate immune response (28944850). Similar results were found in other study applying a systems biology approach, where 217 lncRNA differentially expressed in PBMCs were suggested as potential factors involved in IgA nephropathy pathogenicity (33874912). In this article, HOTAIR was the topmost lncRNA in regulating differentially expressed genes/miRNAs in IgA nephropathy. A recent study determined serum levels of exosomal lncRNA in patients with IgAN and found that the lncRNA-G21551 was down-regulated and may be a potential surrogate biomarker of the disease (32234013)

2.4.4. Membranous Nephropathy

The role of lncRNA has been also explored in membranous nephropathy (MN). Increased levels of the lncRNA XIST were found in urinary samples of MN patients and kidneys from mice with MN (25157805). In cultured podocytes, down-regulation of lncRNA XIST inhibited angiotensin II-mediated apoptosis by suppressing the miR-217-TLR4 pathway (30414341)

For cancer, page 11-12.

2.5. Renal cancers.

As with proteins and miRNAs, lncRNAs can be divided in two categories, cancer-wise: the pro-tumoral (increased in the tumour) and the tumour suppressors (decreased in the tumour). Unsurprisingly when dealing with gene regulators, lots of lncRNA transcripts are associated with tumoral progression such as Xist, one of the best characterized lncRNAs.(2) Xist is located on the long (q) arm of the X chromosome in placental mammals and is instrumental in the X-inactivation process enabling equal expression of the X and autosomal chromosomes. Xist maintains dosage compensation for approximately 1000 genes on the X chromosome, some of which are possible oncogenes. Xist expression has been shown to be decreased in renal cell carcinoma.(98) Additionally, the authors showed that Xist over-expression decreases proliferation and is a stop signal at stage G0/G1 by targeting the miR-106b/P21 axis. The lncRNA ANRIL is overexpressed in several cancers including malignant breast cells, and its subcellular location is an indicator of cancer progression (99). In general, ANRIL overexpression in tumour cells favours proliferation and cell survival, while conversely its inhibition decreases tumour mass and increases apoptosis. The same is true in kidney as ANRIL overexpression favours the invasive potential of renal carcinoma cells by impacting the β-catenin pathway.(100) The roles of MALAT1 RNA, a highly expressed and conserved transcript, in carcinogenesis has been abundantly documented (101). In renal cell carcinoma, MALAT1 is highly expressed and interacts with miR-203 and BIRC5 to increase cell proliferation.(102) These are some of the most striking examples of lncRNAs involved in renal cancer and the reader can refer to for a most comprehensive review.(103)

  1. LncRNAs may be useful as potential biomarkers for the early diagnosis and prognosis of patients with kidney diseases. How about the relative importance of circulatory lncRNA vs. urinary lncRNA? This issue should be discussed.

Answer. We gratefully agree with the reviewer. As a result, we added the following paragraph page 10, in the perspectives section.

At the moment, the main biofluid studied is the blood (as summarized in this review) but urinary expression of lncRNAs has also a strong biomarker potential. At the moment to the best of our knowledge few studies have been published yet in the nephrology field. The study of Khurana et al. that identified urinary exosomal ncRNAs in CKD patients was already discussed in the CKD sub-chapter.(80) A recent study in the related urological field shows that a urinal lncRNA panel has potential clinical value to diagnose bladder cancer.(104) Their smaller counterparts, miRNA were also recently shown to have the same potential in the same organ, but the authors noted that it is important that the controls were age- and sex-related.(105) That will certainly prove true when measuring lncRNAs as well.

  1. Is there any clinical evidences on the function of lncRNA in kidney diseases?

Answer. At the moment, interventional work using lncRNAs were performed only in pre-clinical models and these works were discussed in the manuscript whenever possible. Concerning a potential role of lncRNAs in patients, we discussed this in the Perspectives paragraph

Mini:

   The references in table 1 are missing.

ANSWER: the references were added in the table.

Round 2

Reviewer 2 Report

The authors have addressed my major concerns and the manuscript has been improved. I have no further comments.